# Examination of Upper Extremity Length Discrepancy in Patients with Obstetric Brachial Plexus Paralysis

**DOI:** 10.3390/children10050876

**Published:** 2023-05-13

**Authors:** Murat Danisman, Abdulsamet Emet, Ismail Aykut Kocyigit, Ercan Hassa, Akin Uzumcugil

**Affiliations:** 1Department of Orthopedics and Traumatology, Faculty of Medicine, Giresun University, Giresun 28100, Turkey; murat.danisman@giresun.edu.tr; 2Department of Orthopedics and Traumatology, Etlik City Hospital, Ankara 06170, Turkey; ismail.kocyigit@saglik.gov.tr; 3Department of Orthopedics and Traumatology, Private Memorial Hospital, Ankara 06520, Turkey; dr.ercanhassa@gmail.com; 4Department of Orthopedics and Traumatology, Faculty of Medicine, Hacettepe University, Ankara 06230, Turkey; akinu@hacettepe.edu.tr

**Keywords:** brachial plexus, paralysis, length discrepancy, Narakas

## Abstract

Since the natural course of obstetric brachial plexus palsy is variable, several problems are encountered. One important question, in considering patients with OBPP under observation in outpatient clinical settings, is whether children will have length discrepancies in their arms. The aim of this study was to determine differences in the length of the affected extremity, in comparison to the opposite upper extremity. As such, 45 patients, aged 6 months to 18 years, with unilateral brachial plexus palsy developed due to obstetric reasons, were included in the study. Affected and healthy side humerus, ulna, radius, 2nd metacarpal and 5th metacarpal lengths were evaluated according to gender, age, side, Narakas classification, primary and secondary surgery. Statistically significant differences were found in the change rates of affected/healthy humerus, radius, 2nd metacarpal and 5th metacarpal lengths according to age (93%, 95%, 92%, 90% and 90%, respectively). Affected/healthy change rates of ulna, radius, 2nd metacarpal and 5th metacarpal lengths were found to differ statistically (*p* < 0.05) according to the Narakas classification variable (94%, 92%, 95%, 94% and 94%, respectively). There were no statistically significant differences in the ratios of affected/healthy change in the lengths of the humerus, ulna, radius and 5th metacarpal compared to the primary surgery (*p* > 0.05). The ratios of affected/healthy change in ulna, radius and 5th metacarpal lengths were found to differ statistically (*p* < 0.05) according to secondary surgeries (93%, 91%, 91% and 92%, respectively). Joint and bone deformities and bone shortening were observed after changes that occurred in the postnatal and growing periods due to obstetric brachial plexus palsy. Every increase in function to be gained in the upper extremity musculature was also potentially able to reduce problems, such as shortness.

## 1. Introduction

Obstetric brachial plexus injury (OBPP) is a unilateral or bilateral clinical picture that develops due to injury. It consists of the roots of the brachial plexus, their trunks, their divisions, cords and branches during birth, and is defined by paralysis of various degrees at various levels of the upper extremities [1]. Despite improvements in prenatal and postnatal care, it remains a serious problem. There are several risk factors for this condition, including difficult or prolonged labor, forceps or vacuum extraction during labor, macrosomia, shoulder dystocia and maternal diabetes [2]. In addition, certain positions of the baby during pregnancy, such as breech position or transverse lying, may increase the risk of OBPP. Recent studies mentioned hypotonia as a risk factor, as it is likely to make the brachial plexus more susceptible to stretch [3]. It is important for healthcare professionals to monitor fetal and maternal positions carefully during labor to aid in early detection of potential risks and ensure that appropriate interventions are used to reduce the risk of OBPP. The most commonly-known risk factors have decreased in their considered relationship to OBPP due to continued increases in the rate of cesarean deliveries [4]. In addition, some authors have stated that cesarean delivery has been shown to be protective against OBPP [5]. A recent study showed that, unlike other studies, an increase in birth weight above a recorded minimum was a protective factor against birth-related trauma. In addition, however, the same study posited that cesarean delivery carried risks for the newborn and should be performed only when necessary [6]. More than 50% of babies with OBPP have no known risk factors [7].

The natural course of OBPP has always been the subject of speculation. However, a clear consensus has not yet been reached due to the wide-ranging degrees of involvement. While the definition of recovery has not been made clear, it has been stated that 80–90% complete neural recovery has been seen, according to studies. Meanwhile, other studies have found complete recovery in less that (approximately) 50% of cases [8,9,10,11]. Nervous recovery is not always accompanied by functional recovery, and sequelae, such as joint contractures and subluxations, may remain in some patients [12].

Narakas divided infants with OBPP into 4 groups in 1987 [13]. Narakas I refers to Erb Palsy with C5–C6 involvement. It is the most common form (46%). It represents the patient group with the best prognosis and clinical regression within weeks. Narakas II represents those with C5, C6 ± C7 involvement, which has been seen at a rate of 30%. The prognosis is unpredictable in the first weeks; shoulder function generally does not return, but elbow function can be partially regained. Its prognosis is worse than Narakas I, and it may cause severe symptoms that affect hand function. Narakas III reflects clinical manifestations of avulsion of the upper trunk and C7, stretching of the lower trunk and early postpartum total plexopathy. It has been seen at a rate of 20%. Narakas IV indicates Horner’s sign, which is the most serious form, accompanied by total plexopathy and sympathetic nervous damage.

Diagnosed patients should be placed in treatment as soon as possible due to the importance of early diagnosis and treatment. Serial casting, orthosis usage and functional orthotic applications are some of the methods that have been developed to overcome existing joint contracture [14,15]. However, despite all conservative methods, some permanent sequelae cannot be prevented [16].

Primary procedures, to date, have been based on the principle of neuroma excision and sural nerve grafting for ruptures in the distal foramen [17]. In upper trunk lesions, the main method is to apply nerve grafts from the proximal—that is, from the roots—to the points where neurological conduction cannot be achieved. Another microsurgery method was evaluated, and neurotization of the contralateral C7 with a sural nerve graft to the lower trunk provided significant benefits, especially in terms of hand function, in cases with root-damaged total tumor. In this method, the C7 of the normal side is taken and passed to the opposite side retroesophageally and interposed into the lower trunk [18,19]. This procedure has no known significant comorbidities, apart from mild loss of extension in the unaffected elbow and mild paresthesia of the pollicis and index fingers, especially in adult cases. On the other hand, the procedure has led to significant gains in hand and finger function on the damaged side. In obstetric brachial plexus palsy [20], microsurgical procedures are performed for the sake of functional gain rather than complete recovery [21,22].

Interventions for existing pathoanatomy can be termed secondary procedures [23]. These can be categorized under several headings: relaxation of contractures, tendon transfers and removal of muscle imbalances, joint reduction, nerve decompression and osteotomies [14]. For treatment purposes, many surgical techniques have been described. These techniques can also be used in combination [24].

As the treatment for each patient commences, one important question, considering patients with OBPP under observations in outpatient clinical settings, is whether children will have length discrepancies in their arms. Upper extremity shortness can cause both cosmetic and functional problems. The aim of this study was to determine whether there were differences in the lengths of affected extremities, compared to the opposite upper extremities, in patients with Obstetric Brachial Plexus Paralysis. In addition, the effects of different parameters, such as affected trunk, age and gender, on extremity length were investigated.

## 2. Materials and Methods

The study began after ethics committee approval was obtained from the Hacettepe University Ethics Committee (GO 14/151-03). Between August 2012 and July 2014, patients applied with the diagnosis of obstetric brachial plexus injury—or were evaluated retrospectively—and 45 patients, aged 6 months to 18 years, with unilateral brachial plexus palsy developed due to obstetric reasons, were included in the study. The following were excluded from the study: patients with bilateral plexus damage; patients with plexopathy due to reasons unrelated to obstetrics; patients younger than 6 months or older than 18 years; patients with neurological disease and bone metabolism disease in addition to brachial plexus palsy; patients whose radiographs did not include their intact side or whose radiographs suffered from problems in the PACS system; patients who could not be measured objectively or whose measurements were not taken with a metallic ruler; and patients who were found to have an incorrect position on their radiographs. Affected and healthy side humerus, ulna, radius, 2nd metacarpal and 5th metacarpal lengths were evaluated according to gender, age, side, Narakas classification, primary and secondary surgery.

X-rays were taken in the Department of Radiology by the same technician, under the supervision of an orthopedic surgeon, and using a single X-ray device which also measured length. As with standard-length radiography techniques, each patient’s shoulder was in neutral rotation. Likewise, elbows were in extension, arms were in supination, wrists were in neutral position, and fingers were in abduction. The recordings were made by keeping the tube at a distance of 100 cm from the stand, and the radiopaque metal ruler was placed next to the extremities. The patients included in the study were selected from patients who were able to show the most proximal point of the humerus on the arm and the most distal parts of the fingers on the hand. X-rays were evaluated over the measurement section in the PACS system, and humerus, ulna, radius, 2nd metacarpal and 5th metacarpal length measurements were made on both upper extremities in centimeters (cm) (Figure 1). Measurements were performed by the same orthopedic surgeon, who was blinded to the patients. Humerus measurements were made by taking the most proximal point of the humeral head proximally in patients with and without closure of the physis. The midpoint of the most distal part of the distal humeral condyles adjacent to the joint was taken distally. Radius, ulna and metacarpal measurements were made by taking the middle of the most proximal point of the bone and the middle of the most distal point in patients with closed physis, while the midpoints of the proximal and distal physique level were taken in patients with open physis. While making the measurements, a centimeter was compared with the PACS measurement system on a metallic ruler, which was placed while taking the graph. The measurements were corrected over the smooth coefficients if necessary.

The IBM SPSS Statistics 21.0 package program was used to analyze the obtained data. The Kolmogorov–Smirnov test was used to determine whether the variables to be used in statistical tests fit the normal distribution. Since all variables fit the normal distribution, statistical tests to be used for comparisons were selected from parametric methods. The t test (Paired-t) was used in dependent groups to determine whether the lengths of the humerus, ulna, radius, 2nd metacarpal and 5th metacarpal showed significant differences, according to the affected and healthy arms. In addition, t test (Independent-t) was used in independent groups to determine whether the affected/healthy change ratios of humerus, ulna, radius, 2nd metacarpal and 5th metacarpal lengths differed significantly, according to gender, side, primary and shoulder variables. One-way analysis of variance (One-Way ANOVA) testing was performed to determine whether there were significant differences according to age and Narakas variables. The tests applied were based on a *p* = 0.05 significance level.

There were differences between the bone lengths of the age groups included in the study as the natural influence of age on bone length. In the tests performed for all bone groups, it was observed that the differences in length were statistically significant. For this reason, bone length changes were recorded as % (percentage of difference) instead of cm in order to make the analyses for Narakas, primary and secondary surgery variables more reliable.

## 3. Results

The mean age of 45 patients included in the study was 5.5 ± 3.6 years. The distribution, according to gender, age, Narakas and side variables, is given in the Table 1. The values of the test results, performed to determine whether the lengths of the humerus, ulna, radius, 2nd metacarpal and 5th metacarpal showed significant differences according to the affected and healthy arms, are given in Table 2 and Figure 2. Humerus, ulna, radius, 2nd metacarpal and 5th metacarpal lengths were found to be statistically significantly shorter on the affected side compared to the healthy side (*p* < 0.05). When the affected/healthy percentage differences of humerus, ulna, radius, 2nd metacarpal and 5th metacarpal lengths were evaluated, the parameters did not differ statistically according to gender (*p* > 0.05). The percentage differences of affected/healthy ulna lengths did not show statistically significant differences according to age (*p* > 0.05). A statistically significant difference was found in the percentage differences of affected/healthy humerus, radius, 2nd metacarpal and 5th metacarpal lengths according to age (*p* < 0.05).

The percentage differences of affected/healthy humeral lengths did not show a statistically significant difference according to Narakas classification (*p* > 0.05). Affected/healthy percentage differences of ulna, radius, 2nd metacarpal and 5th metacarpal lengths were found to differ statistically significantly (*p* < 0.05) according to Narakas classification (Table 3).

There were no statistically significant differences in the ratios of affected/healthy change in the lengths of the humerus, ulna, radius and 5th metacarpal compared to primary surgery (*p* > 0.05). The ratios of affected/healthy change in the lengths of the second metacarpal were found to differ significantly (*p* < 0.05) according to primary surgery (Table 4).

When the second metacarpal was evaluated, the mean percentage difference for those who underwent primary surgery was 81.49%, and the mean percentage difference for those who did not undergo surgery was 91.65%.

The ratio of affected/healthy changes in humerus and second metacarpal lengths did not show statistically significant differences according to secondary surgery (*p* > 0.05). The ratio of affected/healthy changes in ulna, radius and 5th metacarpal lengths was found to differ statistically significantly (*p* < 0.05) according to secondary surgery (Table 5).

When the ulna was evaluated, the mean percentage difference for those who underwent secondary surgery was 91.95%, and the mean percentage difference for those who did not undergo surgery was 87.56%. When the radius was evaluated, the mean percentage difference for those who underwent secondary surgery was 93.57% and the mean percentage difference for those who did not undergo surgery was 88.81%. When the 5th metacarpal was evaluated, the mean percentage difference for those who underwent secondary surgery was 92.16%, and the mean percentage difference for those who did not undergo surgery was 83.45%.

## 4. Discussion

Clinicians have long observed that children with obstetric brachial plexus injuries have a shorter and smaller affected upper extremity. The results of the study showed that there was an observable shortness on the side of the affected extremity. Humerus, ulna, radius, 2nd and 5th metacarpal bones of the affected side were found to be significantly shorter compared to the healthy side in patients with OBPP. As expected, no correlation was found between side and gender and shortness. The percentage of shortness gradually increased with age. However, in this study, we observed a correlation between the severity of the injury and the effects of secondary surgeries on extremity length. In the Narakas classification, the amount of shortening in the radius and ulna, especially in the metacarpals, increased as the stage progressed. As the Narakas stage increased, total plexopathy involving the lower trunks occurred. Generally, in total plexopathies, while the upper roots are ruptured, the lower roots are avulsed. Due to this further damage to the lower roots, it can be said that the forearm region, representing C8–T1, is more affected by shortness. Since the upper trunk is involved in every stage of the Narakas classification, the result of the humerus not being affected can be applied to this situation.

According to this study, the degree of shortness was not found to be significantly different in patients who underwent primary surgery. Despite the expectation that muscle denervation would decrease after nerve transfer, it was notable that no changes in shortness were observed. The orthopedic literature considers bone growth primarily dependent on bone loading, but the relationship between the nerves and the skeletal system has been known, to a certain extent, since the 19th century [25,26,27]. In most studies executed, bone elongation was interrupted after denervation of the nerves that innervated the surrounding muscle tissues attached to the long bones. It has been shown that paralyzed limbs are shorter and have less muscle mass in polio patients [28]. Similarly, it was noted that the affected side extremities of patients with hemiplegic cerebral palsy were shorter than the healthy extremities [29]. As suggested, nervous injury not only causes denervation, but also alters the function of the muscles. Therefore, weakened or paralyzed muscles reduce bone load. Then, depending on prolonged paralysis in patients, the muscles undergo atrophy and fatty degeneration, and weak or absent muscle functions occur that impair growth [26,30,31]. Obstetric brachial plexus palsy can also be considered a partial denervation. Similar to the hypotheses put forward, local neuropeptide release due to denervation may be decreased or lost [32]. Recovery of nerve denervation of the bones or partial recovery of muscle function may not be sufficient. It is conceivable that these atrophic muscles cannot produce the healthy mechanical stress that is thought to be necessary for longitudinal growth.

In OBPP, the possibility of deformity, contracture and dysplasia as a result of asymmetric muscle forces increases if secondary surgery is delayed, especially in patients in whom reinnervation cannot restore normal function [33]. The surgeon must consider the limb length difference when making a surgical decision. When the literature was evaluated, the importance of the affected upper extremity appearance was subjectively measured in a family of 48 children with obstetric brachial plexus injury. As the outcome of the study, families rated it extremely important, with an average score of 2.4 on a four-point scale [34]. Clinicians also observed that a smaller limb caused a significant lack of self-confidence in school-aged children and adolescents.

These results supported the hypothesis that untreated obstetric brachial plexus injury could have a sustained, progressive adverse effect on limb growth—and that surgical intervention for obstetric brachial plexus injury could improve limb growth potential. Shortness of more than 5 cm, which may develop due to deficiencies in treatment modalities, can reduce the quality of life, both cosmetically and functionally [35,36]. In the treatment of this limb shortness, treatments such as the Ilizarov method, which are used in the treatment of lower extremity shortness, could be applied. However, many complications and difficulties, such as muscle contractures, joint subluxations, vascular and neurological damage, early consolidation, delayed union, nonunion, refracture, pin infection and psychological impact, should serve as reminders to seek prevention, rather than treatment, of shortness. 

Untreated lower extremity length discrepancies cause significant functional limitations, e.g., pelvic tilt, scoliosis and lower extremity joint problems. Upper extremity length differences due to OBBP can be expected to cause less significant functional limitations, due to the fact that the upper extremity is not a load-bearing limb. Nonetheless, these differences cause social and psychological problems in both families and children due to the cosmetic effect created. The fact that a functionally less-active limb is also cosmetically problematic may cause children with OBPP to be exposed to peer bullying, especially during adolescence, and may lead to social withdrawal or social isolation by reducing the child’s self-confidence. It should be noted that this may also have impacts on the families of these children. In order to evaluate the cosmetic effects of upper extremity length differences due to OBPP, prom studies involving patients and parents must be undertaken.

There were some limitations in this study. The retrospective nature of the study was an important limitation. Moreover, due to the small number of patients and their lack of maturity, lengths could not be fully evaluated. We did not have a control group that had never had surgery; due to the nature of this disease, all patients had undergone surgery. In addition, the total functional results of the patients were not available in this study.

## 5. Conclusions

In conclusion, due to obstetric brachial plexus palsy, limb length discrepancies can be expected after the changes that occur in the postnatal and growth periods. Every increase in the function of the upper extremity muscles reduces problems such as shortness. The family should be informed about prognosis. Therefore, treatment planning for obstetric brachial plexus palsy should be done carefully.

## Figures and Tables

**Figure 1 children-10-00876-f001:**
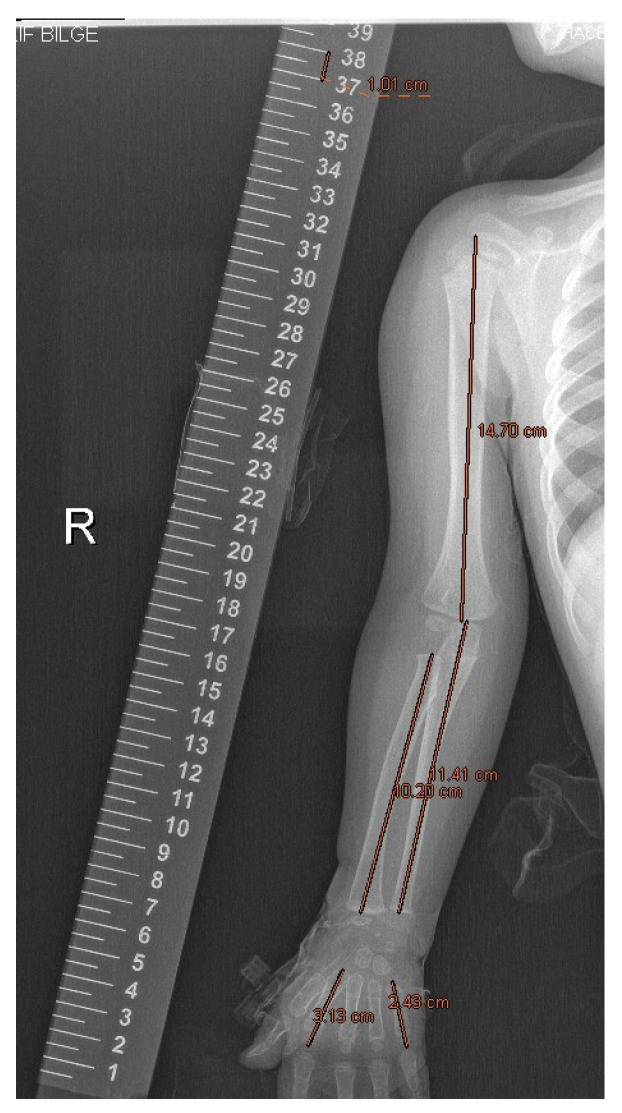
Measurement of upper extremity.

**Figure 2 children-10-00876-f002:**
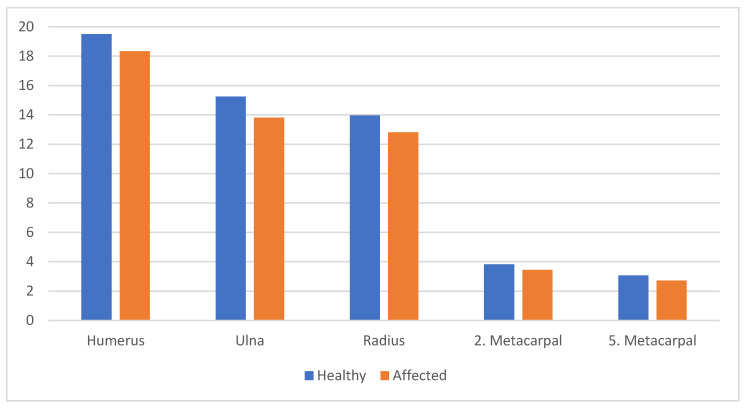
Humerus, ulna, radius, 2nd metacarpal and 5th metacarpal lengths.

**Table 1 children-10-00876-t001:** Distribution of patients according to parameters.

		Number	Percentage (%)
Gender	Male	23	51.1
	Female	22	48.9
Age	0–4	21	46.7
	4–8	14	31.1
	≥8	10	22.2
Narakas	2	21	46.7
	3	10	22.2
	4	14	31.1
Side	Right	33	73.3
	Left	12	26.7

**Table 2 children-10-00876-t002:** Comparison of affected and healthy side bone length averages.

		Mean	Number	SD	t	D	*p* Value
Humerus	Affected	18.34	45	5.55	−8.470	44	<0.001
	Healthy	19.51	45	6.20			
Ulna	Affected	13.82	45	4.31	−7.932	44	<0.001
	Healthy	15.25	45	4.89			
Radius	Affected	12.81	45	3.98	−6.047	44	<0.001
	Healthy	13.97	45	4.74			
2. metacarpal	Affected	3.44	39	1.02	−5.886	38	<0.001
	Healthy	3.83	39	1.11			
5. metacarpal	Affected	2.73	40	0.84	−5.864	39	<0.001
	Healthy	3.04	40	0.92			

**Table 3 children-10-00876-t003:** Comparison of affected and healthy side according to Narakas Classification.

Narakas	Humerus			Ulna			Radius			2. Metacarpal			5. Metacarpal		
	2	3	4	2	3	4	2	3	4	2	3	4	2	3	4
Mean (%)	94.97	94.22	93.96	92.93	91.63	88.21	95.39	92.38	88.98	94.46	88.87	83.48	94.56	89.79	83.44
N	21	14	10	21	14	10	21	14	10	20	9	10	20	9	11
SD	0.037	0.024	0.034	0.051	0.037	0.051	0.051	0.034	0.065	0.086	0.065	0.084	0.083	0.091	0.086
t/F	0.412			4.06			6.183			6.246			5.975		
*p* value	0.665			0.024			0.004			0.005			0.006		

**Table 4 children-10-00876-t004:** Statistical analysis by primary surgery parameter.

Primary Surgery	Humerus		Ulna		Radius		2. Metacarpal		5. Metacarpal	
	+	−	+	−	+	−	+	−	+	−
Mean (%)	94.41	94.5	87.5	91.74	91.05	92.98	81.49	91.65	85.31	90.99
N	6	39	6	39	6	39	5	34	4	36
SD	0.027	0.034	0.024	0.052	0.02	0.062	0.072	0.088	0.075	0.097
t/F	−0.066		−1.933		−0.744		−2.454		−1.125	
*p* value	0.948		0.06		0.461		0.019		0.268	

**Table 5 children-10-00876-t005:** Statistical analysis by secondary surgery parameter.

Secondary Surgeries	Humerus		Ulna		Radius		2. Metacarpal		5. Metacarpal	
	+	−	+	−	+	−	+	−	+	−
Mean (%)	94.63	93.84	91.95	87.56	93.57	88.81	91.48	85.18	92.16	83.45
N	37	8	37	8	37	8	32	7	32	8
SD	0.035	0.026	0.051	0.042	0.054	0.065	0.094	0.061	0.091	0.088
t/F	0.595		2.291		2.163		1.682		2.432	
*p* value	0.555		0.027		0.036		0.101		0.02	

## Data Availability

The data used and/or analyzed during the current study are available from the corresponding author upon reasonable request.

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
