# Peer review of "Examination of Upper Extremity Length Discrepancy in Patients with Obstetric Brachial Plexus Paralysis"

_children, 2023, doi:10.3390/children10050876_

Round 1

Reviewer 1 Report

The paper presents a study on 45 individuals with OBPP and a wide patient age range (6 months to 18 years), investigating length discrepancies of the upper extremity bones in relation to prognostic factors such as age and gender. In general, the paper provides interesting data but there are some shortcomings that require improvement. 

The abstract should include absolute data as well and should be revised according to improved statistical methods.

The introduction can be written more concise.

In materials and methods, please indicate who assessed the radiographs, and were they blinded? In addition, please consult a statistician; analysis with multiple factors requires a different statistical approach.

The results should be revised according to the adjusted statistical approach and preferably presented graphically. What do the authors mean with "rate of change", do they mean left-right percentage difference? This requires further explanation in the methods section.

The discussion should also address the clinical relevance of the small differences found.

The conclusion should really reflect the results of the present study.

Table 2: p values are probable less than 0.001

The English language is fair but can be improved.

Author Response

Dear Editor and Reviewers,

Thank you very much for your valuable comments. Your comments were insightful and helped us to improve our manuscript.

Yours sincerely,

Abdulsamet EMET

Reviewer(s)’ Comments to Author:

Reviewer 1

The paper presents a study on 45 individuals with OBPP and a wide patient age range (6 months to 18 years), investigating length discrepancies of the upper extremity bones in relation to prognostic factors such as age and gender. In general, the paper provides interesting data but there are some shortcomings that require improvement. 

Comments (In order of appearance):

1) The abstract should include absolute data as well and should be revised according to improved statistical methods.

2) The introduction can be written more concise.

3) In materials and methods, please indicate who assessed the radiographs, and were they blinded? In addition, please consult a statistician; analysis with multiple factors requires a different statistical approach.

4) The results should be revised according to the adjusted statistical approach and preferably presented graphically. What do the authors mean with "rate of change", do they mean left-right percentage difference? This requires further explanation in the methods section.

5) The discussion should also address the clinical relevance of the small differences found.

6) The conclusion should really reflect the results of the present study.

7) Table 2: p values are probable less than 0.001

Response to Reviewer 1:

1) Abstract is revised as suggested.

2) As suggested by reviewer 2, new information has been added to introduction. It could not be simplified because it was intended to be more descriptive and there is a lot of information in the literature on this subject. Although it can be written more simply, the recommendations of the publishing journal on the word count of the manuscript were also influential in our decision.

3) In the material and method, information on who evaluated the x-rays is included as suggested. Thank you very much for your comments. Although we have worked with statistician since the plan of the study, we may not have been able to write very clearly and comprehensively while describing our statistical method. The statistics section was rewritten more clearly and in detail.

            First, in the statistical evaluation of our study, the humerus, ulna, radius, 2nd metacarpal and 5th metacarpal lengths of the affected and healthy arms of the same patient were evaluated, and a single measurement was made for the patients, and these measurements naturally vary according to age. For this reason, all statistical calculations were made by using percentage change instead of cm. In order to evaluate the difference in length between the patient and healthy arms, which is the primary target of the study, the “paired t-test” was used as in previous similar studies.

            Secondly, Independent-t and One-Way ANOVA tests with post hoc were applied to test the difference between bone lengths by age, gender, degree of Narakas classification and whether previous surgeries were effective. Apart from these methods, logistic regression analysis is one of the methods that can be done to show the extent to which the same variables affect dependent variables such as humeral length. While there is a clear threshold value of 2 cm when deciding on the surgical treatment of lower extremity length discrepancies due to various diseases, the surgical decision for upper extremity length differences is mostly shaped by cosmetic expectations. Since certain threshold values ​​that we can use in order to categorize our measurements as surgical/non-surgery are not available in the literature, “logistic” regression analysis was not used. Instead, "linear" regression analysis was applied to show the effects of independent variables (age, gender, etc.) on dependent variables (humeral length, etc.). However, as a result of linear regression analysis, it was seen that other independent variables, except age, did not have any effect on the dependent variables. It was thought that this result resulted from the regression analysis because the age distribution in the sample was not homogeneously distributed to the independent variables and the natural influence of age on bone length.

            We discussed with our statistician and we considered that the analysis of data would not cause significant difference in the interpretation of data for our purpose. In order not to cause confusion by using different statistical methods, we did not include the results in manuscript. We would like to present the table of linear regression analysis for your interpretation, if you have additional suggestion in regards of this issue we would give pleasure to hear.

Tables for Linear Regression Analysis:

Humerus

Variable

Coeff.

Std. Err.

p

Gender

M (reference)

0

F

0.618

0.964

0.525

Narakas

2 (reference)

0

3

0.724

1.381

0.603

4

0.820

1.379

0.556

Age

1 (reference)

0

2

-1.687

1.146

0.150

3

-4.658

1.334

0.001

Side

right (reference)

0

left

-0.068

1.137

0.953

Primary

0 (reference)

0

1

-0.358

1.631

0.828

secondary

0 (reference)

0

1

0.807

1.327

0.547

Intercept

94.733

1.586

0.000

Ulna

Variable

Coeff.

Std. Err.

p

Gender

M (reference)

0

F

-0.183

1.401

0.897

Narakas

2 (reference)

0

3

1.815

2.008

0.372

4

-1.338

2.005

0.509

Age

1 (reference)

0

2

-1.077

1.666

0.522

3

-3.969

1.939

0.048

Side

right (reference)

0

left

-1.341

1.653

0.423

primary

0 (reference)

0

1

-3.073

2.372

0.203

secondary

0 (reference)

0

1

4.546

1.929

0.024

Intercept

89.525

2.306

0.000

Radius

Variable

Coeff.

Std. Err.

p

Gender

M (reference)

0

F

-0.501

1.424

0.727

Narakas

2 (reference)

0

3

0.331

2.040

0.872

4

-3.297

2.037

0.114

Age

1 (reference)

0

2

-2.438

1.692

0.158

3

-6.769

1.970

0.002

Side

right (reference)

0

left

-2.096

1.679

0.220

primary

0 (reference)

0

1

0.597

2.409

0.806

secondary

0 (reference)

0

1

3.918

1.959

0.053

Intercept

93.444

2.342

0.000

2. Metacarpal

Variable

Coeff.

Std. Err.

p

Gender

M (reference)

0

F

-2.367

2.294

0.309

Narakas

2 (reference)

0

3

-2.899

3.287

0.384

4

-5.857

3.281

0.083

Age

1 (reference)

0

2

-5.677

2.727

0.044

3

0.647

3.174

0.840

Side

right (reference)

0

left

-1.417

2.706

0.604

primary

0 (reference)

0

1

-4.014

3.882

0.308

secondary

0 (reference)

0

1

3.816

3.157

0.235

Intercept

93.591

3.774

0.000

5. Metacarpal

Variable

Coeff.

Std. Err.

p

gender

E (reference)

0

K

-0.909

2.519

0.720

Narakas

2 (reference)

0

3

-1.843

3.610

0.613

4

-7.282

3.604

0.051

Age

1 (reference)

0

2

-7.284

2.995

0.020

3

-1.481

3.485

0.674

Side

Right (reference)

0

Left

1.077

2.972

0.719

primary

0 (reference)

0

1

-2.308

4.264

0.592

secondary

0 (reference)

0

1

6.304

3.467

0.077

Intercept

90.541

4.145

0.000

4) Further explanation is done in results and new graphic is added as suggested.

5) Clinical relevance is added as suggested.

6) Conclusion have been rewritten to be more clear as suggested.

7) Table 2 is corrected as suggested.

Reviewer 2 Report

Dear Authors,

Thank you for the opportunity to review this paper.

The introduction is elaborate, but are there any known risk factors associated with brachial plexus palsy? Is there any prophylaxis that may be done to diminish the incidence? For example, there are articles that show risk factors in obstetric clavicle fractures, such as this: Obstetric fractures in cesarean delivery and risk factors as evaluated by pediatric surgeons, DOI 10.1007/s00264-022-05547-2. Please elaborate.

Please use a dot instead of a comma in fractions (I.E. 0.037 instead of 0,037).

Please rewrite lines 221-222 for disambiguation.

How many of these patients underwent secondary surgery? Are they all untreated?

Can you introduce a function grading of the affected upper limb? How many of these children underwent full restitution of motor activities?

Conclusions should be rewritten based on your results, to be more clear. I.E. Brachial plexus palsy leads to limb length discrepancies and bone deformities.

Minor English revision is needed.

Author Response

Dear Editor and Reviewers,

Thank you very much for your valuable comments. Your comments were insightful and helped us to improve our manuscript.

Yours sincerely,

Abdulsamet EMET

Reviewer(s)’ Comments to Author:

Reviewer 2

Dear Authors,

Thank you for the opportunity to review this paper.

1) The introduction is elaborate, but are there any known risk factors associated with brachial plexus palsy? Is there any prophylaxis that may be done to diminish the incidence? For example, there are articles that show risk factors in obstetric clavicle fractures, such as this: Obstetric fractures in cesarean delivery and risk factors as evaluated by pediatric surgeons, DOI 10.1007/s00264-022-05547-2. Please elaborate.

2) Please use a dot instead of a comma in fractions (I.E. 0.037 instead of 0,037).

3) Please rewrite lines 221-222 for disambiguation.

4) How many of these patients underwent secondary surgery? Are they all untreated?

5) Can you introduce a function grading of the affected upper limb? How many of these children underwent full restitution of motor activities?

6) Conclusions should be rewritten based on your results, to be more clear. I.E. Brachial plexus palsy leads to limb length discrepancies and bone deformities.

Response to Reviewer 1:

1) Thank you very much for your comments. As you indicated, the risk factors and prophylaxis that may be done to diminish the incidence is added to introduction.

2) All of the commas are corrected.

3) The line rewritten for disambiguation.

4) A total of 37 patients underwent secondary surgery. (Table 5) As mentioned, secondary surgery is performed when patients have existing pathoanatomy. If a positive result is not obtained after primary surgery, relaxation of contractures, tendon transfers and removal of muscle imbalances, joint reduction, nerve decompression and osteotomies  can be performed again on the same patients. Patients do not need a secondary surgery when function is achieved after appropriate nerve repair with sural nerve grafting, neurotization of the contralateral C7, in the follow-ups. Re-surgery or secondary surgery is not planned for patients with successful results, non-surgical treatments are applied to increase function.

5) The primary aim of the study was to evaluate the difference in length between the affected and healthy arms. Unfortunately, the total functional results of the patients are not available in this study. There are many function evaluation methods in the literature. The most frequently used muscle function tool is the Active Movement Scale. Apart from this, Mallet Function Classification (MFS) and Gilbert-Raimondi Evaluation System are also available, with less use. With this seminal recommendation of yours, patients will be called for follow-up again and a functional evaluation will be made in the future.

6) Conclusion have been rewritten to be more clear as suggested.

Round 2

Reviewer 1 Report

Dear authors,

Thank you for revising the paper and providing additional statistical data. In my opinion, the linear regression analysis is the proper method to assess your data and in fact affects your conclusions as well. I would encourage you to completely revise the paper according to the linear regression analysis and leave out your initial statistical approach.

Thank you,

minor editing needed

Author Response

Dear Editor and Reviewers,

Thank you very much for your valuable comments to improve our manuscript.

Yours sincerely,

Abdulsamet EMET

Reviewer(s)’ Comments to Author:

Reviewer 1

Dear authors,

Thank you for revising the paper and providing additional statistical data. In my opinion, the linear regression analysis is the proper method to assess your data and in fact affects your conclusions as well. I would encourage you to completely revise the paper according to the linear regression analysis and leave out your initial statistical approach.

Thank you,

Response to Reviewer 1:

As we mentioned before, we discussed with our statistician and we considered that the analysis of data would not cause significant difference in the interpretation of data for our purpose. We present the table of linear regression analysis for your interpretation. However, we felt the need to discuss these results again with another statistician in line with your suggestions. Our conclusion at the end of our discussion is that Linear Regression Analysis is used for multivariate data analysis. Although there is no problem in performing Linear Regression Analysis in this study, the result will unfortunately be "meaningless". In order to perform linear regression analysis under optimal conditions, the generally accepted rule is ten (10) times the number of variables. The humerus, ulna, radius, 2nd metacarpal and 5th metacarpal variables in the study were 10 in total, including the healthy/affected sides. According to this situation, the number of patients in the study should be 100 in total, but we have 45 patients. For this reason, the conclusion we reached as a result of our consultation with the 2nd statistician is that this analysis will not be suitable for this study again. Reference article: How Many Subjects Does It Take To Do A Regression Analysis. Multivariate Behav Res. 1991 Jul 1;26(3):499-510. DOI: 10.1207/s15327906mbr2603_7.

            The second statistician we consulted suggested that the Kruskal-Wallis test can be used for 3-variable analysis. In the application of this test, no additional changes were detected among the results that we tried to highlight in the study. We did not include the Kruskal-Wallis test results because they were similar. We would like to present the results of the Kruskal-Wallis analysis to your comments, if you wish, and if you have any additional suggestions, we would be happy to hear.

Reviewer 2 Report

Dear Authors,

I appreciate that most of the comments were addressed.

Introduction should include reference to https://link.springer.com/article/10.1007/s00264-022-05547-2 . The recent study suggest that in the 3rd semester a logistic regression model could be used for estimation and prediction probability for an associated pathology to happen, such as OBPP, depending on birth weight and type of delivery.

Such information is mandatory to be used, and thus lowering the risk of OBPP +/- obstetrical fractures, by planning ahead the type of delivery.

Line 51-52: "In addition, cesarean delivery has been shown to be protective for OBPP [5]" - not all literature aproves, and should be changed to "Some authors consider that cesarean delivery has been shown to be protective for OBPP [5]".

We hope that our comments improve your manuscript and ads up quality for the scientific content.

Ok.

Author Response

Dear Editor and Reviewers,

Thank you very much for your valuable comments to improve our manuscript.

Yours sincerely,

Abdulsamet EMET

Reviewer(s)’ Comments to Author:

Reviewer 2

Dear Authors,

I appreciate that most of the comments were addressed.

Introduction should include reference to https://link.springer.com/article/10.1007/s00264-022-05547-2 . The recent study suggest that in the 3rd semester a logistic regression model could be used for estimation and prediction probability for an associated pathology to happen, such as OBPP, depending on birth weight and type of delivery.

Such information is mandatory to be used, and thus lowering the risk of OBPP +/- obstetrical fractures, by planning ahead the type of delivery.

Line 51-52: "In addition, cesarean delivery has been shown to be protective for OBPP [5]" - not all literature aproves, and should be changed to "Some authors consider that cesarean delivery has been shown to be protective for OBPP [5]".

Response to Reviewer 2:

Introduction is corrected as suggested and new reference added.
